# Skilled Performers Show Right Parietal Lateralization during Anticipation of Volleyball Attacks

**DOI:** 10.3390/brainsci13081204

**Published:** 2023-08-15

**Authors:** Brady S. DeCouto, Nicholas J. Smeeton, A. Mark Williams

**Affiliations:** 1Healthspan, Resilience & Performance Research Group, Florida Institute for Human & Machine Cognition, 40 South Alcaniz St., Pensacola, FL 32502, USA; mwilliams@ihmc.org; 2Department of Health & Kinesiology, College of Health, University of Utah, 383 Colorow Drive, Suite 260, Salt Lake City, UT 84112, USA; 3Sport and Exercise Science and Sport Medicine Research and Enterprise Group, School of Sport and Health Sciences, University of Brighton, 1 Denton Road, Brighton BN22 7SR, UK; n.j.smeeton@brighton.ac.uk

**Keywords:** EEG, parietal, Navon, brain, sport, global, local

## Abstract

Global and local biological motion processing are likely influenced by an observer’s perceptual experience. Skilled athletes anticipating an opponent’s movements use globally distributed motion information, while less skilled athletes focus on single kinematic cues. Published reports have demonstrated that attention can be primed globally or locally before perceptual tasks; such an intervention could highlight motion processing mechanisms used by skilled and less skilled observers. In this study, we examined skill differences in biological motion processing using attentional priming. Skilled (*N* = 16) and less skilled (*N* = 16) players anticipated temporally occluded videos of volleyball attacks after being primed using a Navon matching task while parietal EEG was measured. Skilled players were more accurate than less skilled players across priming conditions. Global priming improved performance in both skill groups. Skilled players showed significantly reduced alpha and beta power in the right compared to left parietal region, but brain activity was not affected by the priming interventions. Our findings highlight the importance of right parietal dominance for skilled performers, which may be functional for inhibiting left hemispheric local processing or enhancing visual spatial attention for dynamic visual scenes. Further work is needed to systematically determine the function of this pattern of brain activity during skilled anticipation.

## 1. Introduction

Biological motion perception is impacted by how much experience an individual has with a stimulus [1,2]. For instance, when observing point light displays, newborn infants do not show a preference for coherent human walkers compared to spatially scrambled walkers; yet, with development, these individuals are better at recognizing coherent walking displays [3,4]. The improved recognition of point light walkers can be attributed to a more developed perceptual system that can process spatiotemporal relationships between points/segments, and such improvements are due to perceptual experience [1]. Similarly, skilled athletes can more efficiently process opponents’ motion characteristics compared to novices due to their perceptual experience [5]. To highlight these differences, researchers using spatial occlusion paradigms have demonstrated that skilled athletes use more globally distributed kinematic information when anticipating an opponent’s movements, while less skilled performers use local movement end-effectors (e.g., arm–racquet in tennis) [6,7,8]. While researchers have shown that skilled performers have distinct neural activity from less skilled performers during action perception and use different information cues, few have teased apart skill differences in how motion is being processed.

Published reports suggest that skilled observers use more global processing involving holistic perception of multiple body segments in a movement, while less skilled observers use more local processing involving the perception of single reliable kinematic cues. For example, observers of inverted or scrambled point light displays localize attention on the foot to determine walking direction due to unfamiliarity with the stimulus [9]. In sport, less skilled athletes localize attention on an opponent’s movement end-effector to anticipate an action [10]. Yet, no inferences can be drawn about skill differences in global and local processing as empirical research has not been conducted on the topic. Previously, researchers have not compared between groups of different skill and have predominantly involved the use of point light displays of actions (e.g., walking) with which individuals might have varying levels of experience [9,11]. Sports with complex movement patterns present an ideal domain to investigate the influence of skill on biological motion processing since skilled individuals will possess greater perceptual experience with opponents’ movements than less skilled observers. Thus, the aim of the current investigation is to assess skill differences in how biological motion is processed using skilled and less skilled athletes.

Skill-related differences in motion processing can be examined by priming attention before biological motion processing. Attentional priming biases attention towards the global or local level of a stimulus [12,13]. For tasks where individuals predominantly use a particular processing strategy (e.g., global), priming an incongruent processing strategy (e.g., local) will negatively impact performance. For instance, when individuals verbally describe a face or read the local components of a Navon letter, the later recognition of those faces is impaired, presumably due to a disruption of automated non-verbal facial encoding processes [14,15,16,17]. For face recognition, configural processing (i.e., perceiving facial features in relation to locations of other facial features) is considered superior to more local processing strategies entailing the analysis of individual facial features [18,19]. Thus, when faces are preceded by verbal descriptions or local Navon letters, later recognition is impaired because attention is biased towards local stimulus features during encoding. Similarly, it is plausible that attentional priming impacts biological motion processing. Skilled observers might be most impacted by local priming which would be incongruent with their preference for global processing, and conversely, less skilled perceivers might be most impacted by global priming.

The left hemisphere is considered integral for local processing, making activity in this region an ideal candidate for assessing global and local processing in response to attentional priming. Schooler [13] proposed that verbalization (e.g., describing faces) activates the left hemisphere which is responsible for language and verbal-analytic processing [20]. Similarly, local processing of static images is known to primarily occur in the left hemisphere, and the parietal lobe has particularly been implicated in such processing [21,22,23,24]. Patients with temporoparietal lesions in the left or right hemisphere show impairments in local or global processing, respectively [21,25]. Furthermore, using electroencephalography (EEG), functional magnetic resonance imaging (fMRI), or rhythmic transcranial magnetic stimulation (TMS) researchers have demonstrated advantages in global or local processing according to right or left temporoparietal/parietal activity [22,24,26]. Scientists have also used hand contractions to prime global or local attention, reasoning that unilateral hand contractions can indirectly activate the left or right centroparietal hemisphere for hierarchical stimulus processing [27,28,29].

In sum, the literature broadly suggests that global processing is facilitated by right hemisphere activation, and local processing is facilitated by left hemisphere activation, particularly in the parietal regions [23,30]. Specifically, alpha oscillatory frequencies (8–13 Hz) in the left parietal region are known to facilitate global perception through the inhibition of left-hemispheric local processing functions [22,23,24]. Furthermore, bilateral beta power (14–30 Hz) and greater bilateral beta relative to alpha power before stimulus onset have been associated with facilitated local processing [31,32]. Therefore, activity in left and right parietal regions can index global and local motion processing shifts according to skill level and priming.

Few researchers have looked at how skill impacts neural indices of global or local processing in biological motion displays [33]. In a previous study, our research group assessed how skilled and less skilled soccer players responded to penalty kicks with either global or local motion information degraded. Removing global motion information resulted in higher bilateral beta power relative to alpha power (beta minus alpha) in the parietal cortex, which is thought to enhance visual alertness for detecting local stimulus details [34,35,36,37]. Additionally, we reported decreases in bilateral beta power only in skilled observers when global motion information was removed, suggesting that they were more dependent upon global motion information than less skilled performers. That is, less bilateral beta power likely reflected less top-down attentional control and enhanced parietal activation since skilled performers were less certain about the task outcome without global information [38,39]. However, these findings warrant further investigation given their novelty compared to past literature on the neuroscience of expertise. Therefore, in the present study, we extend our understanding of how parietal alpha and beta power relate to skill and motion processing preferences. Specifically, we elucidate whether skilled and less skilled performers show more neural evidence for global and local motion processing, respectively. To accomplish our aim, we used attentional priming interventions in skilled and less skilled players anticipating attacks in beach volleyball.

We predicted that local priming would elicit increases in bilateral parietal beta relative to alpha power (beta–alpha power) for both skill groups since this is a neural marker of local processing [31]. Moreover, we expected that global priming would induce increases in left parietal alpha power, which is indicative of enhanced global processing [22,24,40]. Regarding skill-related predictions, we expected interactions between skill and priming conditions to emerge, with local priming having larger effects on neural activity in skilled players (i.e., greater increases in parietal beta–alpha power), and global priming having larger effects on neural activity in less skilled players (i.e., greater increases in left parietal alpha power), due to priming conditions activating brain processes not typically utilized by respective skill groups [13]. Furthermore, we expected local priming to reduce bilateral parietal beta power in skilled players only, as per our previous related work [37]. For performance-related outcomes, we hypothesized that local priming would impair anticipation performance in skilled players only since local priming would disrupt their preference for global processing. Additionally, we predicted that global priming would improve anticipation in less skilled players given the previously established facilitative effects of global priming in other domains [41]. Finally, we expected skilled players to demonstrate superior performance compared to less skilled players across priming conditions due to their enhanced perceptual experience with the task [8].

## 2. Materials and Methods

### 2.1. Participants

We recruited 16 skilled (M age = 26.5 ± 4.6, 7 females) and 16 less skilled (M age = 26.4 ± 5.0, 7 females) right-handed beach volleyball players. Of the skilled participants, 6 competed at the AA level, 1 at the AAA level, 8 at the Open level, and 1 at the AVP level (listed by ascending difficulty). Skilled players reported on average 6.7 ± 3.7 years of beach volleyball experience, 8.3 ± 5.5 years of indoor volleyball experience, playing, on average, 7.4 ± 3.4 h of beach volleyball per week. Less skilled participants reported on average 0.3 ± 0.5 years of beach volleyball experience, 0.1 ± 0.3 years of indoor volleyball experience, and 0.1 ± 0.3 h of beach volleyball per week. One skilled participant was removed from the data set because of equipment issues. Additionally, one skilled participant was left-handed. Because of potential lateralization differences due to handedness, we analyzed neural data with and without this participant’s data and achieved statistical results which did not differ. Participants received monetary compensation for their participation in the research. Participants provided informed consent, and all procedures were approved by the lead institution’s Institutional Review Board (IRB).

### 2.2. Stimuli

We recorded videos of 3 AA/Open-level beach volleyball players receiving sets and hitting the ball from the left and right side of the net. The camera was set up in the back corner of the defender’s court in the deep angle position (Figure 1). Players had three possible shots that were categorized by quadrants on the court, namely, short angle, deep angle, or deep line. During each hit, a defender was present at the net blocking the line. The camera was set up at eye-height 170 cm off the ground. Players performed each type of shot 10–15 times from both the left and right sides of the net (a total of 60–90 hits per player). We used a total of 30 different videos from each player with evenly distributed shots, with each video being shown twice (once per condition). Adobe Premiere Pro 2020 (Adobe Systems Incorporated, San Jose, CA, USA) was used to temporally occlude videos at 140 ms and 280 ms before the attacker hit the ball because piloting showed that these intervals showed the greatest differences in performance across skill. We constructed 16 Navon letters using the letters D, E, F, and H with Microsoft PowerPoint 2021 (Microsoft, Redmond, WA, USA). Global letters were 10 cm wide and local letters were 1.3 cm wide, which resulted in a visual angle of 9.22° and 1.17°, respectively, based on a viewing distance of 62 cm. Letters were white presented on a black background in size 14 font. We used PsychoPy v2021.3 to present the stimuli [42]. The computer monitor was 53 × 32 cm with a refresh rate of 60 Hz and pixel resolution of 1920 × 1080 pixels. The computer and monitor were stationed in the corner of a laboratory in which only the participant and experimenter were present. The computer was equipped with an Intel Core i7-7700 K CPU, 32 Gb of RAM, and an NVIDIA Quadro P4000 graphics card.

### 2.3. Procedures

Participants filled out demographic questionnaires, then sat 0.62 m away from a computer monitor while the EEG cap was prepared. First, a baseline period of 1 min resting EEG was collected. Participants were exposed to 6 temporally occluded videos to familiarize them with the task, and feedback was provided on their keyboard responses. During the task, the participants were instructed to minimize movement to reduce motion artifact during data collection. Each video trial consisted of a still image of the first video frame for 500 ms, the attacker receiving and passing a hit to the setter, the setter passing to the attacker, the attacker approaching the ball to swing (~3 s), and a black screen for 2 s during which participants were told to make a keyboard response to decide the ball location. ‘Enter’ signified front right court, ‘right Ctrl’ back right court, ‘Caps Lock’ front left court, and ‘left Ctrl’ back left court. Participants were instructed that the attacker would never hit the ball in the same quadrant as the defender/blocker (e.g., short line shot) and that the defender was always blocking the line. Following the 6 practice trials, participants completed 3 blocks of 30 video trials which served as more practice.

In the subsequent conditions, the participants anticipated volleyball shots with a local or global Navon matching task (counterbalanced) interleaved between videos. The Navon matching task involves two Navon letters presented adjacent to each other, and participants were instructed to determine whether the two objects matched at the global or local level (Figure 1) [43]. Participants pressed ‘Right Ctrl’ for a match and ‘Left Ctrl’ for no match. Left and right keyboard responses were evenly distributed. Navon trials were preceded by a fixation cross (1 s), and the letters remained visible for 3 s. The Navon trials ended when participants made a keyboard response. The participants practiced the Navon matching task for 15 trials, then completed 3 blocks of 30 video trials with a Navon matching task between each video (Figure 2). The participants performed the Navon matching task at the global and local level for a total of 6 blocks of 30 trials. The entire experiment consisted of 180 recorded trials (276 including practice) and took roughly 40 min to complete.

### 2.4. Measurements

**Anticipation.** Performance was measured by comparing each participant’s response for the ball location relative to the actual ball location. Performance was analyzed as the percentage (%) of trials answered correctly.

**Brain Activity.** EEG data were processed using BrainVision Recorder and BrainVision Analzyer 2 (Brain Products, GmbH, Munich, Germany). We used 32 channels of an actiCAP system with electrodes labeled in accordance with a 10–20 system [44]. Signals were amplified with a BrainAmp DC amplifier. The system collects data at 1000 Hz, and impedances at each electrode were kept below 25 kΩ. The ground electrode was placed on the right earlobe, and the reference electrode was placed directly anterior to Cz. We used prefrontal (Fp1, Fp2) and frontotemporal (FT9, FT10) electrodes on the cap as vEOG and hEOG electrodes. We resampled the data to 256 Hz, and an infinite impulse response (IIR) filter was applied with a high-pass filter at 0.1 Hz and a low-pass filter at 60 Hz (4th order). We used independent component analysis (ICA) with vEOG and hEOG to correct for ocular artifact. We removed components that were influenced by ocular activity (e.g., sum of squared correlations > 5) from the data. We transformed the data using surface Laplacian with a 4th order spline. The data were manually inspected for major muscular and blink artifacts that the ICA did not correct.

We applied fast Fourier transformation (FFT) to the data to analyze oscillatory rhythms in the alpha (8–13 Hz) and beta (14–30 Hz) frequency ranges. Additionally, we analyzed the difference between beta and alpha power (beta minus alpha power, or beta–alpha) because this measure has been associated with local processing bias [32]. We specifically examined activity in the left (P3 and P4 average) and right (P7 and P8 average) parietal regions. Alpha and beta oscillations were measured during two different epochs. The first epoch included 500 ms of the still image preceding each video (preparatory period), and the second epoch included the last 800 ms of each video. This 800 ms interval was selected because it included the attackers’ approach, jump, and swing, with minimal information from factors extraneous to motion processing contributing to the final ball location (e.g., movements from the setter and the beginning of the ball trajectory did not influence ball-strike outcomes). We subtracted each participant’s FFT data from their 60 s baseline period.

### 2.5. Data Analysis

The behavioral (*W* = 0.987, *p* = 0.081) and EEG data (Alpha: *W* = 0.983, *p* = 0.022; Beta: *W* = 0.973, *p* = 0.001) were log10-transformed to obtain a more normal distribution. In assessing performance during the Navon matching task, one participant failed to follow instructions for the global priming condition; thus, we removed these data from the global priming condition. For all analyses, we used linear mixed effect regressions (LMERs) to control for individual differences in intercepts within each condition because random effects can control for individual responses to conditions [45,46,47]. Performance models included factors of skill (skilled, less skilled), occlusion (140 ms, 280 ms), condition (local prime, global prime), interaction terms, and random effects of participant:Performance ~ Skill × Condition × Occlusion + (1|Participant)

We used separate LMERs for each epoch to analyze the EEG data, with factors of skill (skilled, less skilled), hemisphere (right, left), condition (local prime, global prime), interaction terms, and random effects of participant and participant crossed with condition and hemisphere:EEG ~ Skill × Hemisphere × Condition + (1|Participant) + (1|Hemisphere:Participant)
+ (1|Condition:Participant)

The effect sizes for main models were calculated as partial eta-squared values (*η*^2^), and effect sizes for post hoc comparisons were calculated as Cohen’s *d* values. Multiple testing corrections were applied to post hoc tests using Benjamini–Hochberg *p*-value corrections. For significant effects involving factors with two levels, we report beta estimates and standard errors alongside ANOVA statistics [48]. All statistical procedures were conducted in R Studio using R v4.1.3 [49,50,51].

## 3. Results

### 3.1. Volleyball Task Performance

The full model results for the performance model are presented in Table 1. There were significant main effects for skill (*β* = 0.072, *SE* = 0.018, *p* < 0.001, *η*^2^ = 0.167), condition (*β* = 0.022, *SE* = 0.011, *p* = 0.041, *η*^2^ = 0.051), and occlusion (*β* = 0.134, *SE* = 0.016, *p* < 0.001, *η*^2^ = 0.459). Skilled players performed significantly better than less skilled performers across conditions. Across skill groupings, performance was significantly better in the global priming compared to local priming condition. Finally, performance was significantly better in the 140 ms occlusion interval than in the 240 ms interval. No significant interactions emerged (*p*’s > 0.300). The results are presented in Figure 3.

### 3.2. Brain Activity

*Alpha Power.* The full model results are presented in Table 2 and Table 3. During the preparatory period, there was a significant interaction between skill and hemisphere (*p* = 0.002, *η*^2^ = 0.228). There was lower alpha power in the right compared to left hemisphere for skilled players (*p* < 0.001, *d* = 0.83) but not for less skilled players (*p* = 0.852, *d* = 0.03). Additionally, less skilled players showed greater alpha power in the right hemisphere than skilled players (*p* = 0.032, *d* = 0.44). No significant effects emerged for condition (*p* = 0.891, *η*^2^ = 0.001), and no interactions emerged between skill and condition (*p* = 0.624, *η*^2^ = 0.006) or hemisphere and condition (*p* = 0.271, *η*^2^ = 0.032).

During the motion processing period, there was a significant interaction between skill and hemisphere (*p* = 0.001, *η*^2^ = 0.250). Lower alpha power was evident in the right compared to left hemisphere for skilled players (*p* < 0.001, *d* = 1.10), but not for less skilled players (*p* = 0.453, *d* = 0.17) (see Figure 4). No significant effects emerged for condition (*p* = 0.678, *η*^2^ = 0.004), or interactions between skill and condition (*p* = 0.075, *η*^2^ = 0.074), and hemisphere and condition (*p* = 0.697, *η*^2^ = 0.004).

*Beta Power.* The full model results are presented in Table 2 and Table 3. During the preparatory period, there was an interaction between skill and hemisphere (*p* = 0.024, *η*^2^ = 0.127). There was lower beta power in the right compared to left hemisphere for skilled (*p* = 0.001, *d* = 1.03) and less skilled players (*p* = 0.022, *d* = 0.45), but this hemispheric difference was significantly larger for skilled players (*β = 0*.210, *SE* = 0.088, *p* = 0.023, *d* = 0.30). No significant effects emerged for condition (*p* = 0.090, *η*^2^ = 0.073) nor interactions between skill and condition (*p* = 0.899, *η*^2^ < 0.001), or hemisphere and condition (*p* = 0.736, *η*^2^ = 0.003).

During the motion processing period, there was a significant interaction between skill and hemisphere, (*p* = 0.022, *η*^2^ = 0.129). There was lower beta power in the right compared to left hemisphere for skilled (*p* < 0.001, *d* = 1.03) and less skilled players (*p* = 0.036, *d* = 0.43), but this hemispheric difference was significantly larger for skilled players (*β = 0*.219, *SE* = 0.089, *p* = 0.019, *d* = 0.31) (Figure 4). No significant effects emerged for condition (*p* = 0.306, *η*^2^ = 0.027) or interactions between skill and condition (*p* = 0.515, *η*^2^ = 0.011), and hemisphere and condition (*p* = 0.591, *η*^2^ = 0.007).

*Beta–Alpha Power*. The full model results are presented in Table 2 and Table 3. During the preparatory period, there was a significant main effect for hemisphere (*p* = 0.001, *η*^2^ = 0.288). The left hemisphere showed greater beta–alpha power for both skill groups across conditions (*p* = 0.001, *d* = 0.46). There was no main effect of skill (*p* = 0.337, *η*^2^ = 0.026), condition (*p* = 0.214, *η*^2^ = 0.014), or significant interactions between skill and hemisphere (*p* = 0.452, *η*^2^ = 0.016), skill and condition (*p* = 0.792, *η*^2^ = 0.002), or condition and hemisphere (*p* = 0.483, *η*^2^ = 0.014).

During the motion processing period, there was a significant main effect for hemisphere, (*p* = 0.007, *η*^2^ = 0.182), revealing that across skill groupings and conditions, there was greater beta–alpha power in the left parietal region compared to right parietal region (*p* = 0.009, *d* = 0.50). There was no main effect for skill (*p* = 0.967, *η*^2^ < 0.001), condition (*p* = 0.307, *η*^2^ = 0.028), or interactions between skill and hemisphere (*p* = 0.568, *η*^2^ = 0.009), skill and condition (*p* = 0.742, *η*^2^ = 0.003), or hemisphere and condition (*p* = 0.263, *η*^2^ = 0.034).

## 4. Discussion

We examined skill differences in motion processing by priming global and local attention in skilled and less skilled players while they anticipated filmed volleyball attacks. We expected local priming to adversely impact only skilled performance and induce less bilateral beta power, whereas global priming was predicted to positively influence only less skilled performance and elicit greater shifts towards global processing through greater left parietal alpha power [24]. Finally, we predicted that skilled performers would generally show more lateralized alpha power in the left hemisphere to inhibit local processing [22,24].

In contrast to our predictions, players in both skill groups performed better after global priming compared to local priming. Furthermore, priming conditions did not elicit distinct neural activity. Notably, skilled performers had significantly reduced beta and alpha power in the right compared to left parietal region across conditions, which reflects increased activation of the right parietal region [52,53,54]. Moreover, only skilled players showed reduced alpha power in the right hemisphere, further demonstrating that they rely more on right parietal functions for accurate information extraction during anticipation. These skill-based neural adaptations to the anticipation task may be functional for global processing which is associated with right hemispheric activation [23]. However, the lack of neural effects in response to global and local priming leaves other possible explanations open for discussion regarding right parietal function. Furthermore, such hemispheric asymmetry has not been observed in the limited work on skill-related anticipation [38,55,56]. Considering our results alongside previous studies on the neuroscience of expertise, the observed hemispheric asymmetry in skilled players provides novel insight into what factors within a sporting domain influence parietal lobe activation.

### 4.1. Right Parietal Lateralization in Skilled Performers

Thus far, few researchers have measured EEG while participants anticipate an opponent’s actions [56]. Denis and colleagues conducted a study investigating skill-related differences in brain activity during the anticipation of tennis serves [55]. Their results showed that experienced athletes had greater ERD for beta and mu oscillations in sensorimotor brain regions, both of which are correlates of enhanced mirror system activation; however, experienced athletes did not show hemispheric differences in parietal alpha power as evidenced in the current paper [55]. These differences may be explained by several factors. Denis et al. analyzed clusters based on independent components rather than measuring brain activity at parietal electrodes. Furthermore, we applied surface Laplacian to our data to improve spatial resolution of electrode signals. Such analysis choices that were appropriate for our study design can partly explain differences between our findings and those of Denis et al.’s. However, in our previous investigation on skilled and less-skilled soccer players using similar EEG processing procedures, hemispheric asymmetries in alpha and beta power were not present in skilled performers anticipating penalty kicks [37]. Given the large effect sizes for alpha asymmetry in the present experiment, it is likely that the observed asymmetry is related specifically to the visual constraints of the volleyball stimuli. That is, the number of players and movement of the ball in the volleyball task may require distinct neural resources to extract relevant information from the display compared to the soccer task, which included only a single opponent and a stationary ball. The work of Del Percio and et al. lends support to this notion, since only experienced soccer players observing visual scenes with multiple players showed greater reductions in bilateral parietal alpha power [57]. The soccer task employed by Del Percio et al. did not involve biological motion processing, but rather distance judgements between players. Therefore, for our volleyball task, alpha reductions specific to the right hemisphere may be functional for task-specific goals such as biological motion processing.

Given the priming conditions did not elicit further shifts in hemispheric asymmetry as expected, the right hemispheric dominance exhibited by skilled performers could be indicative of other cognitive functions besides global motion processing. Similar trends have been noted in skilled marksmen who have lower alpha and beta power in the right compared to left hemisphere than novices during the aiming period before shooting [58,59,60,61]. Greater alpha and beta power in the left hemisphere is interpreted as functional for the inhibition of left hemispheric verbal-analytic functions, and lower alpha and beta power in right hemisphere reflects the activation of visual spatial functions, which are more pertinent for successful task performance [58,62]. Skilled performance is generally marked by automaticity during motor execution; thus, left hemispheric inhibition can reduce conscious regulation of task-related processes that could otherwise disrupt task performance [58,63]. Although our task involves biological motion processing rather than self-paced motor execution (e.g., shooting), the parietal asymmetry observed in skilled volleyball players may overlap with previous evidence from studies involving marksmen. Skilled players may have routine perceptual strategies for identifying what information they should extract during the time-course of an opponent’s movements, and these processes likely occur without much conscious effort [64]. In contrast, the perceptual strategies used by less skilled observers might be less automatic, and consequently, they may use a more consciously regulated strategy as evidenced by less right parietal engagement.

In the context of past work on the neuroscience of expertise, our findings add insight to the frequently studied action observation network (AON) that is involved when individuals observe an action [65,66]. Researchers using fMRI have shown that observing and anticipating familiar actions results in more robust activation of AON regions, including the cerebellum, superior parietal lobe, and intraparietal sulcus [66]. Specifically, the superior parietal lobe is conjectured to be more active for skilled action observation because of its role in providing domain-specific contextual information during anticipation [65,66]. While our study used EEG, the observed increase in right parietal activation lends some support to work on the AON. Moreover, published reports provide support for the “neural efficiency hypothesis” (NEH), which asserts that skilled observers can recruit relevant neural resources more efficiently than novice observers to anticipate actions, entailing reduced cortical activation [56]. For example, Babiloni and colleagues showed that expert karate athletes observing videos of karate actions demonstrated more pronounced alpha event-related desynchronization (ERD) than non-athletes only in brain regions most relevant to task completion, namely the fronto-parietal and mirror pathways [67]. Thus, our results align with the NEH, showing that skilled players recruit brain regions that are most pertinent to successful task completion (i.e., right parietal region for our experimental task).

Although we selected an epoch which highlighted only the attacker’s approach and striking action, there may have been more visual information to filter out to optimally focus on the volleyball attacker, which skilled players could do most effectively. Thus, greater right parietal activation could reflect more advanced visual spatial processing in the presence of more visual distractors. Furthermore, greater beta–alpha power was evident in the left parietal region across skill groups and conditions, indicating that our task may have generally required left hemispheric functions to localize attention on the attacker [32,36]. Visual crowding studies, which require local processing of target letters surrounded by distractor letters, show that higher beta and lower alpha power in parietal–occipital regions are associated with improved target letter detection [68,69,70]. In our study, heightened vigilance promoted by left parietal beta–alpha power may have been necessary for optimal focus on the attacker amid surrounding player movements. That is, like visual crowding studies where high beta and low alpha power can improve the detection of a target letter surrounded by distractors, greater left parietal beta–alpha power may have been functional for enhancing a localized attention on the attacker while inhibiting surrounding “distractors” or information sources [69,70].

In sum, the greater right parietal engagement exhibited by skilled performers could reflect an enhanced ability to integrate multiple information sources in the scene. Moreover, the greater left parietal beta–alpha power across skill levels suggests that our task required attention to be more consciously directed towards the most relevant information source (i.e., the attacker) in the presence of distracting information. Along with this conjecture, we would expect that more players on the court (e.g., 4 v 4 scenarios) would elicit greater left-hemispheric beta–alpha power and right parietal dominance because there are more distractors (e.g., opponents/teammates) surrounding the attacker. Further research is necessary to clearly define the relationship between parietal beta and alpha activity in filtering visual information in complex scenes.

### 4.2. Preparatory Period Versus Motion Processing Period

Generally, our neural results from the preparatory period mirror those observed during the motion processing period. In our previous investigation on soccer penalty kicks, only the motion processing period, and not the preparatory period, elicited shifts in neural activity according to viewing conditions that were normal, blurred, or spatially occluded [37]. The preparatory period reflects neural resources being recruited to process an upcoming static stimulus [23,24]. However, motion perception is a continuous process occurring over a discrete time interval; therefore, measuring EEG during this motion processing period is potentially more critical than a preparatory period for assessing underlying neural mechanisms responsible for global or local motion perception. One possible reason why there were no differences between the neural activity elicited in the preparatory period in the present study could be that stimuli were not visually manipulated (i.e., blurred or spatially occluded); thus, the selection of a processing strategy during the preparatory period may have been easier to anticipate from viewing a still image that was familiar to their normal visual experience. In contrast, our previous soccer work featured manipulated visual displays (blurred or spatially occluded), making it difficult for players to select a processing strategy during the preparatory period in the absence of motion [37]. In line with this conjecture, we expect that visually altering the display would wash out similarities between the preparatory and motion processing period in the current paper.

### 4.3. Superior Performance with Global Priming

Our behavioral results showed priming-specific effects. Performance was superior in the global priming condition across skill groupings. Previous reports have shown that local priming negatively impacts skilled performance [41], but our study shows that local priming impaired performance for both skill groups. Global priming may have expanded attention towards more kinematic features for global processing. However, an alternate explanation for the benefits of global priming on performance is that local priming disrupted preferred motion processing strategies [13,15]. Motion perception may be a relatively automatic process, and local priming might have activated verbal-analytic cognitive processes causing perceptual strategies to become more consciously regulated. Performers may have reinvested attention into their perceptual performance akin to how anxious performers might detrimentally reinvest attention on their movements [63,71,72]. Explicitly monitoring movement or perceptual strategies generally results in impaired performance because well-learned perceptual-motor strategies are best performed without conscious interference [73,74,75]. To support these conjectures, more work is needed that directly assesses how reinvestment and conscious regulation impact parietal activity during anticipation and motion perception. Our results lend some support to the notion that interventions promoting global attention may be able to accelerate learning rates for perceptual skill acquisition if global attention is facilitative of task performance. Future investigations on neural activity in different brain regions in response to priming would elucidate these conjectures since the parietal region did not show priming-specific effects.

### 4.4. Limitations

The lack of a similar control condition to the priming conditions is a limitation in the current paper. However, a ‘neutral’ cognitive task that does not use global or local processing would be needed to make valid comparisons for neural activity, and there is no precedence for such tasks. That is, priming likely activates specific neural processes (i.e., for global and local processing), and a ‘neutral’ control task might inadvertently activate global or local processing if not systematically tested carefully. Thus, given our primary interest in priming was comparing global versus local processing, using only the priming conditions is justified but has its limitations. More ecologically valid designs have been shown to better tease out skill differences (e.g., live sport scenarios) [76,77]; therefore, skill differences in neural activity may have been more salient in live settings requiring more realistic motor responses. Finally, in future, researchers should use designs that include neighboring brain regions around the parietal lobe to add insight into how the observed parietal activity relates to global or local motion processing [21,24,32].

## 5. Conclusions

While our findings did not clearly implicate a role for parietal activity in relation to global and local motion processing, the results demonstrate the importance of right parietal dominance during skilled anticipation of dynamic scenes and offer groundwork for researchers to investigate the precise role of such neural patterns in skilled observers for different contexts. The enhanced activity in the right parietal region could facilitate global processing, enhanced visual spatial attention to focus on multiple stimuli, or more automatically regulated perceptual strategies in complex visual scenes with multiple players [38,59,62]. The detrimental effects of local priming on performance could indicate that the activation of verbal-analytic processes disrupts automatic perceptual performance [74]. Alternatively, global priming may have broadened attentional strategy, which would be beneficial for biological motion processing during anticipation. In future, scientists can build upon these findings by systematically manipulating the complexity of visual scenes during anticipation to determine the precise contribution to parietal lateralization during visual attention for skilled performers. Furthermore, different neuroimaging techniques (e.g., TMS) could elucidate how skill-related alpha/beta lateralization extend into other more or less complex domains and scenarios.

## Figures and Tables

**Figure 1 brainsci-13-01204-f001:**
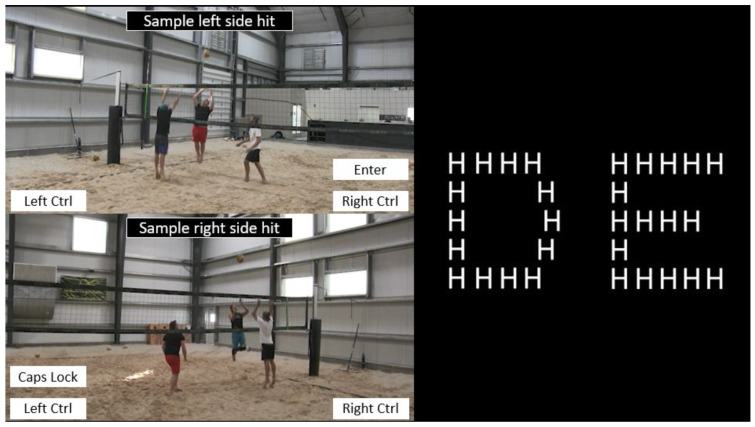
(**Left**) Sample footage of video stimuli. Attackers received a set and hit from either the left or right side of the net. There were four possible keyboard responses that corresponded to the anticipated ball location: front left (‘Caps Lock’) front right (‘Enter’), back left (‘Left Ctrl’), and back right (‘Right Ctrl’). (**Right**) Experimental Navon matching task example. Individuals had to determine whether the two objects matched at the global or local level. In this example, the objects only match at the local level because both are comprise smaller letter “H’s”.

**Figure 2 brainsci-13-01204-f002:**
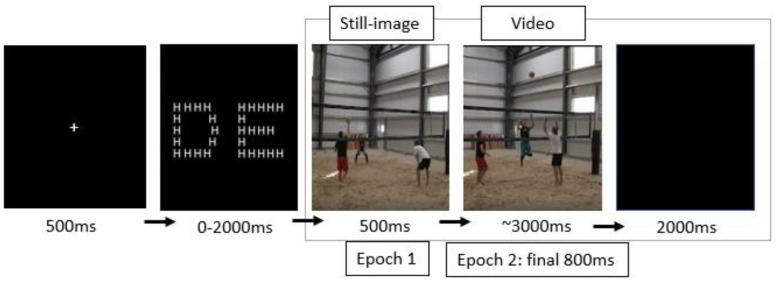
A schematic of the experimental design. The participants first viewed a fixation cross which transitioned to a screen showing the Navon matching task, where individuals were instructed to determine whether or not the two Navon letters matched at the global level or local level (in this figure, Navon letters only match at the local level because they both contain smaller letter h’s). Following the matching task, the volleyball anticipation task was presented. EEG data were collected during the still-image period (epoch 1) and during the final 800 ms of the motion processing period (epoch 2).

**Figure 3 brainsci-13-01204-f003:**
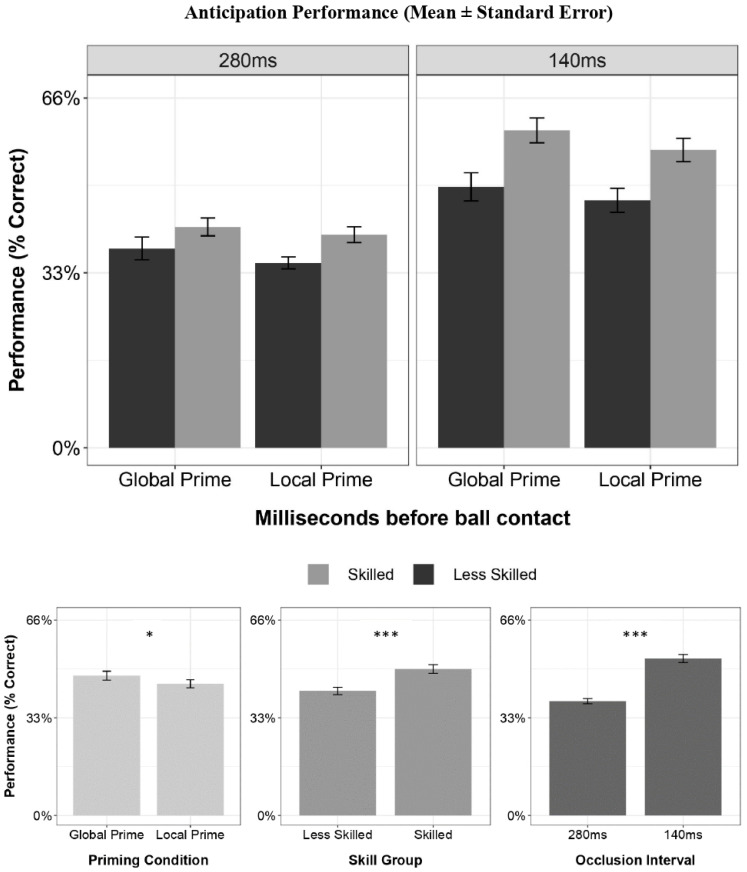
The anticipation accuracy scores for each skill group, condition, and temporal occlusion interval. Global priming elicited significantly greater accuracy than local priming across all factors. Skilled players performed significantly better than less-skilled players across occlusion intervals and conditions. Players performed significantly better in the 140 ms occlusion interval compared to the 240 ms occlusion interval. * *p* < 0.05, *** *p* < 0.001.

**Figure 4 brainsci-13-01204-f004:**
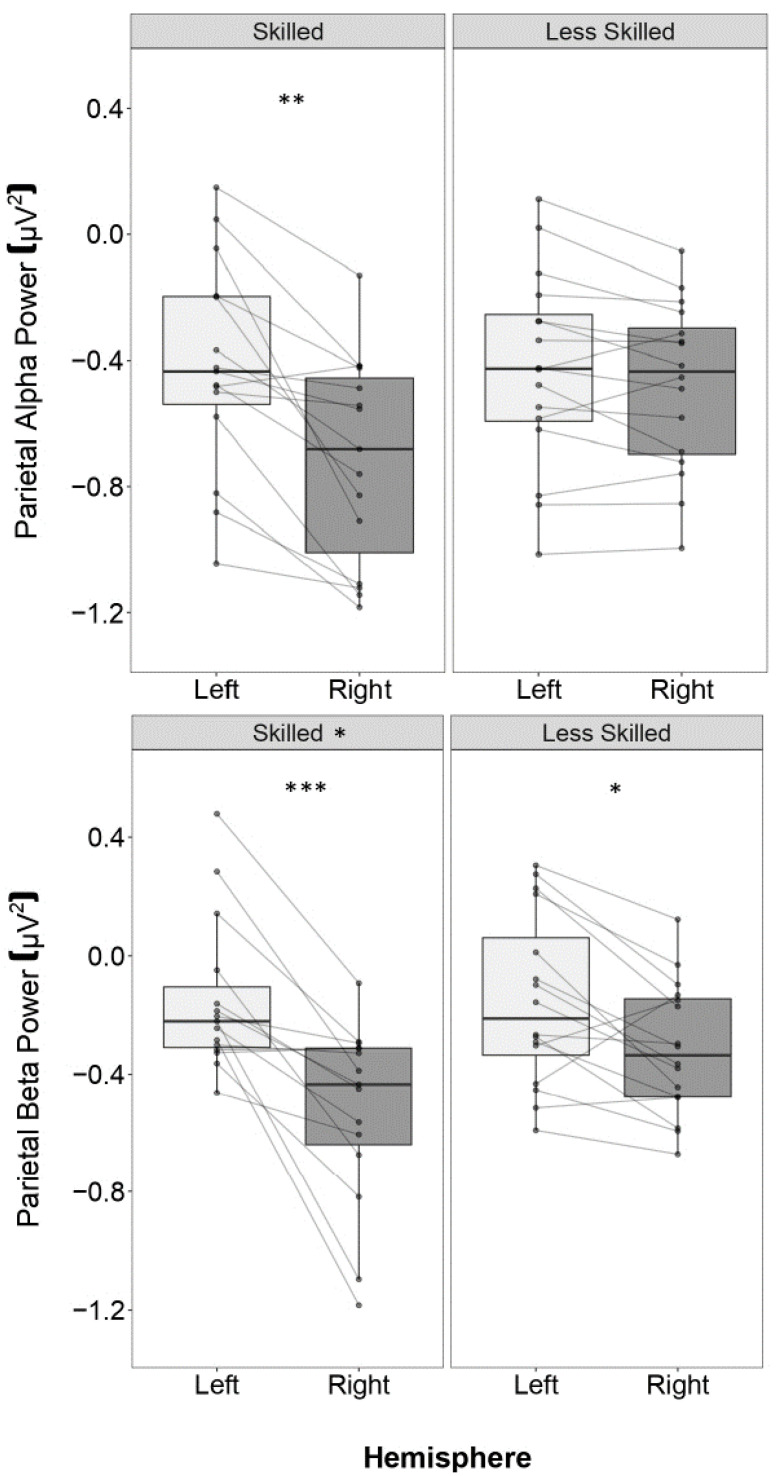
Box plot of log-transformed parietal beta power for each hemisphere and skill group during motion processing. Horizontal lines in boxes represents the median, boxes represent the interquartile range (between Q1 and Q3), and vertical lines represent 1.5x the interquartile range. Individual data points and lines represent each participant’s alpha or beta power differences between hemispheres. Negative values reflect desynchronization from a baseline period. **Top**: Only skilled players demonstrated significant differences in alpha power between hemispheres. **Bottom**: Both skill groups showed significantly reduced beta power in the right compared to left hemisphere, but skilled participants demonstrated larger hemispheric differences. * *p* < 0.05, ** *p* < 0.01, *** *p* < 0.001.

**Table 1 brainsci-13-01204-t001:** Mixed model results for performance analyses. Significant results are highlighted in **bold**.

DV	Fixed Effect	*df*	*F*	*p*	*η* ^2^	*R* ^2^
Performance	**Skill**	**1**, **32**	**17.02**	**<0.001**	**0.167**	0.504
	**Condition**	**1**, **32**	**4.56**	**0.041**	**0.051**	
	**Occlusion**	**1**, **32**	**72.15**	**<0.001**	**0.459**	
	Skill × Condition	1, 32	<0.01	0.950	<0.001	
	Skill × Occlusion	1, 32	1.11	0.300	0.013	
	Condition × Occlusion	1, 31	0.02	0.894	<0.001	
	Skill × Condition × Hemisphere	1, 31	0.27	0.608	0.003	

**Table 2 brainsci-13-01204-t002:** Mixed model results for analyses on EEG measurements. Significant results are highlighted in **bold**.

DV	Epoch	Fixed Effect	*df*	*F*	*p*	*η* ^2^	*R* ^2^
Alpha Power	1	Skill	1, 31	0.96	0.336	0.025	0.184
		**Hemisphere**	**1**, **31**	**13.05**	**0.001**	**0.257**	
		Condition	1, 30	0.02	0.891	0.001	
		**Skill × Hemisphere**	**1**, **31**	**11.18**	**0.002**	**0.228**	
		Skill × Condition	1, 30	0.25	0.624	0.006	
		Condition × Hemisphere	1, 30	1.26	0.271	0.032	
		Skill × Condition × Hemisphere	1, 30	1.04	0.316	0.027	
	2	Skill	1, 31	1.43	0.240	0.033	0.147
		**Hemisphere**	**1**, **31**	**26.58**	**<0.001**	**0.385**	
		Condition	1, 30	0.18	0.678	0.004	
		**Skill × Hemisphere**	**1**, **31**	**14.15**	**0.001**	**0.250**	
		Skill × Condition	1, 30	3.41	0.075	0.074	
		Condition × Hemisphere	1, 30	0.15	0.697	0.004	
		Skill × Condition × Hemisphere	1, 30	1.57	0.220	0.036	
Beta Power	1	Skill	1, 31	2.73	0.108	0.066	0.304
		**Hemisphere**	**1**, **31**	**36.16**	**<0.001**	**0.481**	
		Condition	1, 30	3.07	0.090	0.073	
		**Skill × Hemisphere**	**1**, **31**	**5.67**	**0.024**	**0.127**	
		Skill × Condition	1, 30	0.02	0.899	<0.001	
		Condition × Hemisphere	1, 30	0.12	0.736	0.003	
		Skill × Condition × Hemisphere	1, 30	0.06	0.807	0.002	
	2	Skill	1, 31	2.00	0.167	0.048	0.241
		**Hemisphere**	**1**, **31**	**35.14**	**<0.001**	**0.472**	
		Condition	1, 30	1.09	0.306	0.027	
		**Skill × Hemisphere**	**1**, **31**	**5.81**	**0.022**	**0.129**	
		Skill × Condition	1, 30	0.43	0.515	0.011	
		Condition × Hemisphere	1, 30	0.30	0.591	0.007	
		Skill × Condition × Hemisphere	1, 30	0.28	0.603	0.007	
Beta–Alpha Power	1	Skill	1, 31	0.95	0.337	0.026	0.128
		**Hemisphere**	**1**, **31**	**14.25**	**0.001**	**0.288**	
		Condition	1, 30	1.61	0.214	0.044	
		Skill × Hemisphere	1, 31	0.58	0.452	0.016	
		Skill × Condition	1, 30	0.07	0.792	0.002	
		Condition × Hemisphere	1, 30	0.50	0.483	0.014	
		Skill × Condition × Hemisphere	1, 30	0.51	0.482	0.014	
	2	Skill	1, 31	<0.01	0.967	<0.001	0.046
		**Hemisphere**	**1**, **31**	**8.30**	**0.007**	**0.182**	
		Condition	1, 30	1.08	0.307	0.028	
		Skill × Hemisphere	1, 31	0.33	0.568	0.009	
		Skill × Condition	1, 30	0.11	0.742	0.003	
		Condition × Hemisphere	1, 30	1.30	0.263	0.034	
		Skill × Condition × Hemisphere	1, 30	0.12	0.729	0.003	

**Table 3 brainsci-13-01204-t003:** Post hoc comparisons for EEG mixed models. Significant results are highlighted in **bold**. “Effect/Interaction” refers to the significant main effect or interaction found in the mixed model. “Factor Levels” refers to levels within a factor in which post hoc comparisons are being made for interactions. “*β*” refers to the estimated difference between compared groups. “≈” denotes nonsignificant difference.

Dependent Variable	Effect/Interaction	Factor Levels	Post Hoc Comparison	*β*	*SE*	*p*	*d*
Alpha Power Epoch 1	Skill × Hemisphere	Left Hemisphere	Skilled ≈ Less Skilled	0.066	0.077	0.520	0.15
		**Right Hemisphere**	**Skilled < Less Skilled**	**0.190**	**0.077**	**0.032**	**0.44**
		**Skilled**	**Left H > Right H**	**0.267**	**0.057**	**<0.001**	**0.83**
		Less Skilled	Left H ≈ Right H	0.010	0.055	0.852	0.03
Alpha Power Epoch 2	Skill × Hemisphere	Left Hemisphere	Skilled ≈ Less Skilled	0.005	0.107	0.960	0.01
		Right Hemisphere	Skilled ≈ Less Skilled	0.241	0.107	0.060	0.40
		**Skilled**	**Left H > Right H**	**0.292**	**0.049**	**<0.001**	**1.10**
		Less Skilled	Left H ≈ Right H	0.046	0.047	0.453	0.17
Beta Power Epoch 1	Skill × Hemisphere	Left Hemisphere	Skilled ≈ Less Skilled	0.011	0.084	0.903	0.02
		**Right Hemisphere**	**Skilled < Less Skilled**	**0.218**	**0.084**	**0.022**	**0.46**
		Skilled	Left H > Right H	0.365	0.065	**<0.001**	1.03
		Less Skilled	Left H ≈ Right H	0.158	0.063	**0.022**	0.45
Beta Power Epoch 2	Skill × Hemisphere	Left Hemisphere	Skilled ≈ Less Skilled	0.015	0.099	0.879	0.03
		**Right Hemisphere**	**Skilled < Less Skilled**	**0.226**	**0.099**	**0.036**	**0.41**
		**Skilled**	**Left H > Right H**	**0.364**	**0.065**	**<0.001**	**1.03**
		**Less Skilled**	**Left H > Right H**	**0.154**	**0.063**	**0.036**	**0.43**
Beta–Alpha Power Epoch 1	Hemisphere		**Left H > Right H**	**0.123**	**0.034**	**0.001**	**0.46**
Beta–Alpha Power Epoch 2	Hemisphere		**Left H > Right H**	**0.090**	**0.032**	**0.009**	**0.50**

## Data Availability

Data are publicly available at https://github.com/bradydecouto/GlobalLocalVolleyball (accessed on 10 August 2023).

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
