# Peer review of "Skilled Performers Show Right Parietal Lateralization during Anticipation of Volleyball Attacks"

_brainsci, 2023, doi:10.3390/brainsci13081204_

Round 1

Reviewer 1 Report

The paper presents an interesting exploration into the neural underpinnings of skill and priming effects in motion processing, using EEG measurements. The authors delve into the impacts of global and local priming on proficient and less-experienced volleyball players, as well as the neural activity linked to these effects. The study provides valuable understanding of the relationship between cognitive processes, skill level, and motion perception. However, to fortify the paper, there are essential points to address and recommendations for improvement.

1.     Hypotheses and Methodological Considerations Need Greater Clarity:

a.     The specific hypotheses and research questions are not articulated clearly enough in the paper. It's uncertain whether all participants had the same level of video game experience. For example, a right-handed beach volleyball player with ample experience in motion-based video games might outperform others despite having less skill. Many young people excel at online tennis games, even though they may have little to no actual tennis experience. It seems there were no such outliers in the data presented in the paper. This type of information was not mentioned in section 2.1. However, these circumstances and the inclusion of more specific hypotheses are necessary to corroborate the findings of this report.

2.     Figures and Tables:

a.     It's commendable that the authors offer a range of test administration methods. However, it's unclear why some results are bolded while many others are reported without emphasis. It's challenging to link the data presented in the main text to the table entries. You could highlight or bold all reported items so that readers can follow the sequence and read the table and text concurrently.

b.     The resolutions of plots are too low. When saving plots in R, you can export them with higher resolution as eps files.

c.     The middle panel of Figure 3, bottom, should be labeled with ** (two stars, not three).

d.     A suitable legend is needed for plot 3. The addition of labels such as "mean+/-s.e.m." is necessary.

e.     If your goal is to compare skilled versus less-skilled, it would be more informative to place the plots of less-skilled and skilled side by side. For instance, in the top panel of Figure 3, you could plot 280 ms of less skilled versus 280 ms of skilled, followed by 140 ms of less skilled versus 140 ms of skilled. In the 280ms measurement, there seems to be little difference between the less-skilled and proficient groups.

f.       What does the y-label "? (V2)" in Figure 4 represent?

g.     Minor: In line 277, p=.001 should replace p=001

3.     Discussion:

a.     Comparison with other neural activity data: The paper would benefit from a more comprehensive discussion of the results in relation to previous statistical literature on neural activities. How does the current study connect to earlier research on changes in neural group activity, such as the mean and variance in firing, dynamic changes in neural activity space, and submanifold rotations?

Reviewer 2 Report

Thank you for submitting your work. The context is adequately presented and the methods are clearly mentioned. However, 

  1. The introduction section of the article could benefit from being more concise and to-the-point.

  2. In addition, please mention the aims in accordance with the outcomes of the current research in the introduction.

  3. Furthermore, some references that are not directly related to the current work should be removed.

  4. Please fix the units in the y-axis label in Figure 4.
